# Prostate cancer survivors with symptoms of radiation cystitis have elevated fibrotic and vascular proteins in urine

**Bernadette M. M. Zwaans**[1,2], **Heinz E. Nicolai**[3,4], **Michael B. Chancellor**[1,2], **Laura E. Lamb**[1,2]*

**1** Department of Urology, William Beaumont Hospital, Royal Oak, Michigan, United States of America, **2** Oakland University William Beaumont School of Medicine, Rochester, Michigan, United States of America, **3** Departamento de Urología, Universidad de Chile, Santiago, Chile, **4** Hospital Clínico San Borja Arriarán, Santiago, Chile

* Laura.Lamb@beaumont.org

**Data Availability Statement:** All relevant data are within the manuscript and its Supporting Information files.

## Abstract

Radiation for pelvic cancers can result in severe bladder damage and radiation cystitis (RC), which is characterized by chronic inflammation, fibrosis, and vascular damage. RC development is poorly understood because bladder biopsies are difficult to obtain. The goal of this study is to gain understanding of molecular changes that drive radiation-induced cystitis in cancer survivors using urine samples from prostate cancer survivors with history of radiation therapy. 94 urine samples were collected from prostate cancer survivors with (n = 85) and without (n = 9) history of radiation therapy. 15 patients with radiation history were officially diagnosed with radiation cystitis. Levels of 47 different proteins were measured using Multiplex Luminex. Comparisons were made between non-irradiated and irradiated samples, and within irradiated samples based on radiation cystitis diagnosis, symptom scores or hematuria. Statistical analysis was performed using Welch's t-test. In prostate cancer survivors with history of radiation therapy, elevated levels of PAI 1, TIMP1, TIMP2, HGF and VEGF-A were detected in patients that received a radiation cystitis diagnosis. These proteins were also increased in patients suffering from hematuria or high symptom scores. No inflammatory proteins were detected in the urine, except in patients with gross hematuria and end stage radiation cystitis. Active fibrosis and vascular distress is detectable in the urine through elevated levels of associated proteins. Inflammation is only detected in urine of patients with end-stage radiation cystitis disease. These results suggest that fibrosis and vascular damage drive the development of radiation cystitis and could lead to the development of more targeted treatments.

## Introduction

Approximately 3.6 million prostate cancer (PCa) survivors are currently living in the United States, representing 1/5[th] of all cancer survivors [1]. It is estimated that, in the United States, 191,390 new patients will receive a PCa diagnosis in 2020. Most of these PCa are diagnosed at

**Funding:** This study was supported by the U Can-Cer Vive Foundation (https://ucancervive.com/; BMMZ, LEL, MBC), and NIDDK K01 Career development award DK114334 (https://www.niddk.nih.gov/; BMMZ). The funders had no role in study design, data collection and analysis, decision to publish, or preparation of the manuscript.

**Competing interests:** The authors have declared that no competing interests exist.

an early stage, which contributes to a very high overall 5-year survival rate (99%) [2]. Treatment of PCa is dependent on stage at diagnosis, risk of reoccurrence, patient's age, presence of comorbidities and patient's personal preference, and includes active surveillance, surgery, and radiation, chemotherapy, hormone, and androgen-deprivation therapies [1]. Despite the high survival rate after PCa, many PCa survivors struggle with short or long-term side effects from their cancer therapy. Approximately half the PCa survivors that received radiation therapy or surgery suffer from urinary or bowel dysfunction [1]. Of the PCa survivors that were treated with radiation therapy, a small percentage (range 4–11%) will develop radiation (hemorrhagic) cystitis (RC) that increases with years after radiation therapy [3].

RC is a chronic bladder condition that is characterized by urinary frequency, nocturia, incontinence, pelvic pain and hematuria. These symptoms are caused by severe bladder fibrosis and vascular damage for which no cure exists to date. Treatments are generally focused on arresting bleeding and include blood transfusions, cystoscopy with fulguration/clot evacuation, hyperbaric oxygen therapy, instillation of aminocaproic acid, alum, silver nitrate or formalin, and cystectomy with or without neobladder reconstruction [3–5]. RC treatments have limited effectiveness, are time consuming and can have serious side effects. Thus, there is a dire need for improved treatment options for patients suffering from RC.

In addition to the lack of safe and effective treatments, RC is often diagnosed at a late stage when damage may be irreversible, making it more challenging to effectively treat. Lack of mechanistic insight into the disease progression is in part due to lack of bladder biopsies available for analysis; taking bladder biopsies from RC patients is risky as it may cause additional damage to an already fragile tissue. However, insight into molecular changes after irradiation and RC diagnosis is essential to identify patients early on that are at increased risk for developing RC and to develop targeted treatments. To circumvent the inability to obtain tissue biopsies, we used urine samples of PCa survivors as 'liquid biopsies' to identify altered protein levels. In urological research, urinary biomarkers are widely considered for carcinoma in situ, bladder cancer and interstitial cystitis, but are novel for RC [6, 7].

RC bladders are believed to have three main histological characteristics: accumulation of tissue fibrosis, vascular damage as evidenced by hematuria, and chronic inflammation. Therefore, in this study, we used urine samples from PCa survivors with a history of radiation therapy to identify changes in excreted urinary proteins involved in fibrosis, inflammation and vascular biology.

## Materials and methods

### Urine and data collection

The study was conducted with full Beaumont Institutional Review Board (IRB) approval (#2015–302 and #2018–179). Study participants were recruited at Hospital Clínico San Borja Arriarán, Santiago de Chile, Chile, with approval of the Chilean Comité Ético-Científico (CEC-SSMC # 96/16). Patient population included in the study are Chilean PCa survivors with and without history of external beam radiation therapy. Prostate cancer survivors that concurred to regular control of their prostate cancer, were invited to participate in the study between March 2016 and March 2020. Of those that received radiation treatment, some patients had been diagnosed with RC. RC diagnosis was determined independently by a urologist through cystoscopy, and defined by a pale bladder mucosa and telangiectasia, with or without ulcers, and a decreased maximal bladder capacity as measured during cystoscopy [8]. Exclusion criteria for participation were a history of chemotherapy, interstitial cystitis, recurrent urinary tract infections, kidney and/or bladder stones, prostatitis, or other bladder disorders (e.g. neurogenic bladder) [8]. Participants voluntarily signed a written informed consent

before filling out a health survey and providing a midstream urine sample. The survey included questions on demographics, smoking status, bladder health, symptom severity, and radiation history. For symptom severity, patients scored the incidence of nocturia, incontinence, hematuria, gross hematuria, urgency, and urogenital spasm on a scale from zero (never) to five (multiple times per day) [9]. Scores for each symptom were added to form the total score (maximum = 30). High symptom score was defined as score $\geq$ 10. All patients with micro- and macrohematuria were studied with cystoscopy to confirm RC and to rule out other causes of hematuria. Urine was collected in sterile urine cups at a random time point (e.g. not first urine of the day) and urine preservative (Norgen Biotek) was immediately added to keep urinary proteins stable for up to one year at room temperature. De-identified urine samples and surveys were shipped to William Beaumont Hospital's Research Institute, Royal Oak, MI. Dipstick analysis was performed on urine samples to rule out urinary tract infections and identify presence of hematuria. Urine samples were spun down for 5 minutes at 700 x g and supernatant was stored in aliquots at -80˚C.

## Multiplex luminex assay

Urine samples were thawed on ice and vortexed prior to use. Only urine samples not previously thawed were used for this study. Fibrotic, inflammatory and vascular proteins (Table 1) were measured using the Milliplex multiplex assay system (EMD Millipore) according to the manufacturer's instructions. Samples were run in duplicate simultaneously with known standards and quality control samples on Bio-Plex 200 System (Bio-Rad). This system was previously used to for development or urine biomarkers for interstitial cystitis [6]. Groups that were compared included non-irradiated versus irradiated samples. Of the samples that received radiation therapy, protein changes in the urine were analyzed based on the presence of RC diagnosis, hematuria, and high symptom score.

**Table 1. Detectability of fibrotic, inflammatory and angiogenic proteins in urine samples of prostate cancer survivors using Luminex assay.**

| | Fully detectable | | | Partially detectable | | | Minimally detectable | | |
|---|---|---|---|---|---|---|---|---|---|
| | Proteins | LLD | Avg | Proteins | LLD | Avg | Proteins | LLD | Avg |
| Fibrotic factors | Cathepsin-D TIMP-1 TIMP-2 | 24 20 49 | 55356 321 1546 | MMP-9 MMP-10 MMP-12 PAI-1 TIMP-3 TIMP-4 | 14.0 27.0 98.0 2.0 98.0 10.0 | 783 15 32 9.4 31 5.7 | MMP-1 MMP-2 MMP-3 MMP-7 MMP-13 TGFb1 | 27 68 146 548 58 9.8 | 0 7.2 0 0 0 0.5 |
| Inflammatory factors | MCP-1 MCSF sVCAM-1 | 3.2 97.6 61 | 572 51221 1129 | sICAM-1 | 24.0 | 256 | Fractalkine GMCSF IL-1a* IL-1b IL-4 IL-6* IL-7 IL-8 IL-10 IL-13 IL-17a MIP-1a* TNFa | 3.2 3.2 3.2 3.2 3.2 3.2 3.2 3.2 3.2 3.2 3.2 3.2 3.2 | 0 0.4 0.8 0.02 1.1 2.1 0 20.9 0.1 0.06 0.01 0.62 0.63 |
| Vascular factors | EGF HB-EGF HGF PlGF VEGF-A | 2.7 1.4 27.4 1.4 13.7 | 1092 16 113 52 168 | Endoglin FGF-2 Leptin PDGF-AA | 27.4 13.7 137 2.0 | 28 2 179 13 | Angiopoetin-2 BMP9 GCSF Follistatin PDGF-AB/BB VEGF-D | 13.7 2.7 13.7 27.4 24 6.9 | 0.9 0.4 0.2 1.4 8.8 0.9 |

* Only detectable in 1–2 RC cases with gross hematuria; LLD = lower limit of detection (pg/ml); Avg = average detected protein level in samples (pg/ml).

## Statistical analysis

Statistical analysis was performed using SPSS 26.0. A one-way ANOVA test followed by a post-hoc two-tailed Welch t-test was performed to determine statistical significance with unequal variance between two populations: non-irradiated versus irradiated groups (Fig 1), or, within irradiated group, patients with versus without RC diagnosis, hematuria or high symptom scores (Figs 2–4). Significance level was set at $p \leq 0.05$. Results are expressed as mean ± standard deviation (SD).

# Results

## Patient demographic

The patient population consists of PCa survivors with (n = 79) and without (n = 9) a history of radiation therapy. Of those that received radiation therapy, 15 patients were diagnosed with RC prior to urine collection. All urine samples were collected between March 2016 and March 2019. Patient demographic data, radiation history and symptom severity are summarized in Table 2. All patient groups were comparable in age, BMI and smoking status. Of the PCa survivors with radiation history, urine samples were collected on average 5.5 years after radiation exposure (4.07 yr for RC patients, and 5.8 yr for non-RC patients). The majority of patients received 36–40 radiation treatments, though dose did not appear to determine a patient's risk for developing RC. Overall, more patients with RC suffered from frequency, urgency, nocturia, hematuria, urinary incontinence and bladder spasm. One non-RC case reported hematuria; this patient was diagnosed with RC after data collection.

## Altered urinary protein profile associated with symptoms of radiation damage

Multiplex Luminex assays were used to measure fibrotic, vascular and inflammatory proteins in human urine samples. Of the proteins tested, 22 proteins were within the detectable range of the Luminex assay in all or at least half (partially detectable) of the urine samples (Table 1).

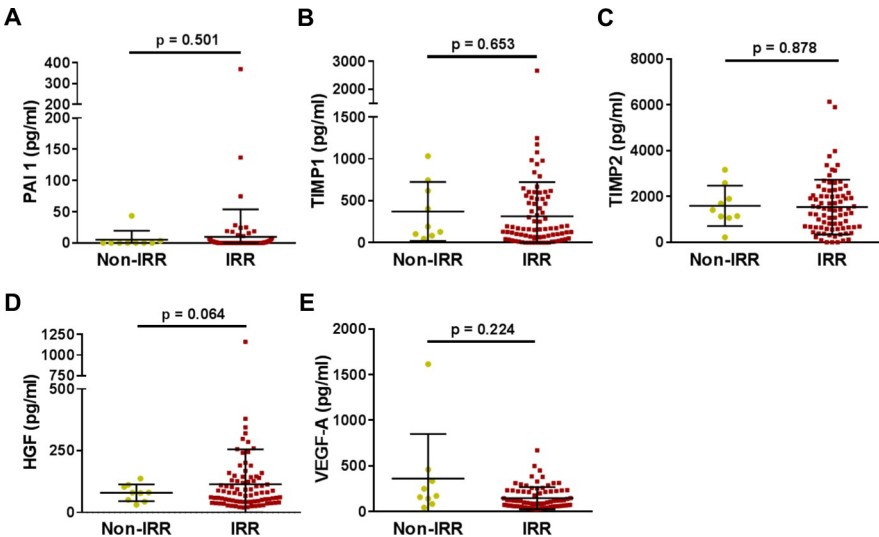

**Fig 1. Fibrotic and angiogenic proteins are not altered in urine after radiation therapy.** (A-C) The fibrotic proteins PAI 1, TIMP1, and TIMP2, and (E) the vascular protein VEGF-A are unaltered between non-irradiated (non-IRR) and irradiated (IRR) PCa survivors. (D) HGF levels are overall higher after irradiation treatment, though this is not significant. Non-IRR: n = 9; IRR: n = 85. Black line = mean; Error bar = SD.

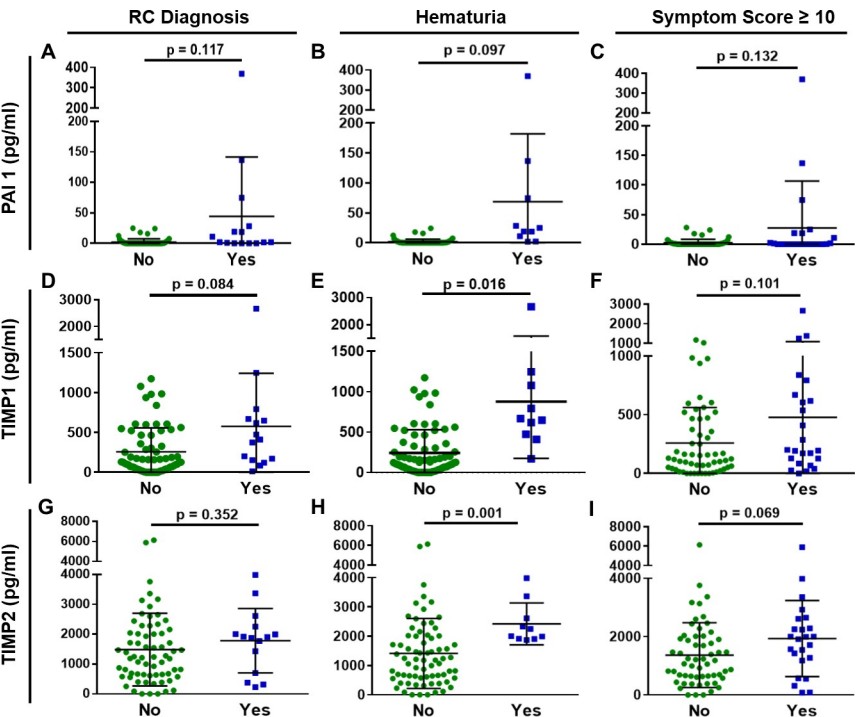

**Fig 2. PAI 1 and TIMP1/2 are altered in urine of prostate cancer survivors.** Increased levels of (A-C) PAI 1, (D-F) TIMP1 and (G-I) TIMP2 levels are measured in PCa survivors with history of radiation treatment with RC diagnosis, hematuria or high symptom score. No: n = 67; Yes: n = 15. Black line = mean; Error bar = SD.

Proteins in the urine were not altered in response to irradiation (Fig 1); only HGF levels, were increased in patients with radiation history, though this did not reach the 5% significance level (Fig 1D).

Within the PCa survivors with a history of radiation therapy, several fibrotic and vascular proteins were altered based on the presence of RC-related symptoms (RC diagnosis, hematuria

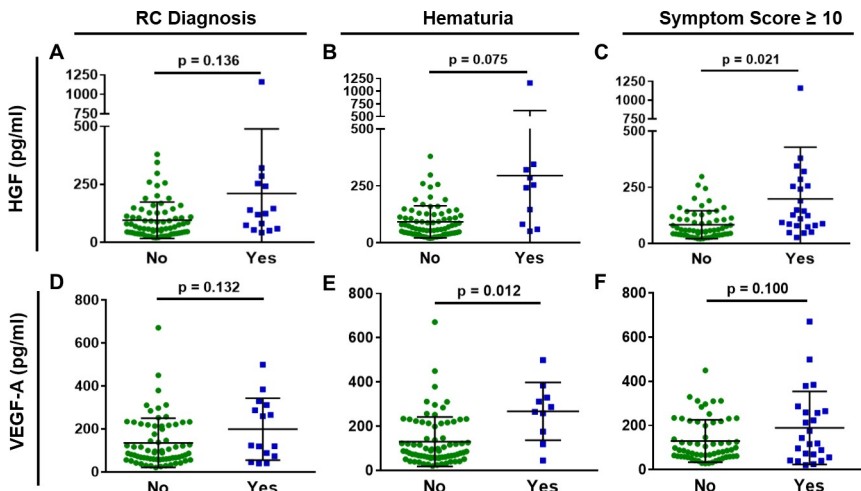

**Fig 3. HGF and VEGF-A levels coincide with the presence of RC symptoms.** Levels of (A-C) HGF and (D-F) VEGF-A are elevated in PCa survivors with RC diagnosis, with hematuria or with a high symptom scores. No: n = 70; Yes: n = 15. Black line = mean; Error bar = SD.

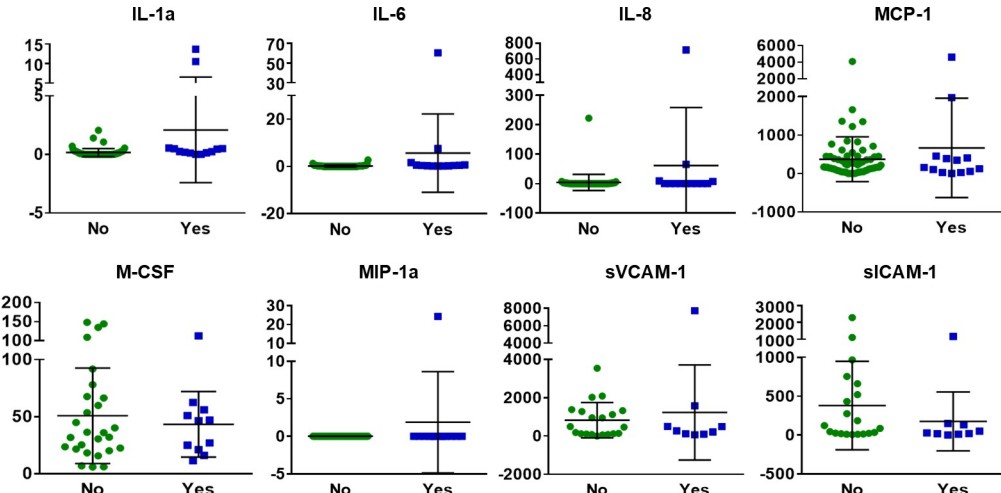

**Fig 4. Urinary inflammatory markers not altered in prostate cancer survivors with RC.** In prostate cancer survivors with radiation history, Inflammatory cytokines were not altered in patients with RC in comparison to patients without RC diagnosis. Y-axis: protein concentration in pg/ml. No: n = 70; Yes: n = 15. Black line = mean; Error bar = SD.

or high symptom score). The fibrotic proteins PAI 1, TIMP1 and TIMP2 were detectable in the urine and their elevated levels were associated with RC diagnosis, hematuria or high symptom score (Fig 2). For PAI 1, its highest increase was seen in samples with hematuria; the three samples with the highest PAI 1 levels were patients with gross hematuria (Fig 2B). TIMP1 and TIMP2 were also elevated in PCa survivors with radiation history and RC symptoms (Fig 2D–2I). TIMP2 levels had a stronger association with high symptom score (Fig 2I), while TIMP1 levels were more closely associated with RC diagnosis (Fig 2D).

The vascular proteins HGF and VEGF-A were also altered in PCa with radiation history based on RC symptoms (Fig 3). HGF protein concentration was significantly higher in patients with high symptom scores and showed a positive association with hematuria and RC diagnosis (Fig 3A–3C). VEGF levels were significantly higher in patients with hematuria versus those that did not report hematuria (Fig 3E). The average VEGF levels were higher for patients with RC diagnosis and patients with high symptom scores, though these lacked statistical significance (Fig 3D–3F).

Of the inflammatory proteins measured, only MCP-1, MCSF and sVCAM-1 were detectable in the urine. However, no association was found between their concentration and RC diagnosis, hematuria or symptom score (Fig 4). IL-1a, IL-6, and MIP-1a were detectable only in 1–2 severe RC cases that reported daily occurrence of gross hematuria. IL-8 was present in 11/94 urine samples, but was not associated with any RC or radiation related symptoms (Fig 4).

## Discussion

RC is characterized in part by fibrosis, hematuria and inflammation. Lack of available bladder biopsies from patients with radiation history makes it challenging to identify pathways that drive the development of RC. As such, targeted therapies cannot be developed, and patients are left with suboptimal treatment options. Thus, we used urine samples as 'liquid bladder biopsies' to look for protein changes in PCa survivors with a history of radiation therapy. We previously identified three elevated vascular markers in the urine of PCa survivors with a history of radiation therapy [8]. We expanded this cohort and looked for changes in inflammatory and fibrotic proteins in addition to vascular proteins (Table 3).

**Table 2. Patient demographics, radiation history and symptom severity.**

| | | Participants with radiation treatment history | | |
| --- | --- | --- | --- | --- |
| | Control | All | RC diagnosis | |
| Total | (n = 9) | (n = 85) | Yes (n = 15) | No (n = 70) |
| Age (y)* | 73.33 (± 9.72) | 72.67 (± 7.39) | 72.94 (± 9.69) | 72.94 (± 6.85) |
| BMI (kg/m$^2$)* | 26.40 (± 4.36) | 27.19 (± 3.30) | 27.29 (± 3.04) | 27.17 (± 3.37) |
| Smoking Status | n = 9 | n = 83 | n = 13 | n = 70 |
| Never | 66.7% (n = 6) | 44.6% (n = 37) | 38.5% (n = 5) | 45.7% (n = 32) |
| Past | 33.3% (n = 3) | 41.0% (n = 34) | 53.8% (n = 7) | 38.6% (n = 27) |
| Yes | 0 | 14.4% (n = 12) | 7.7% (n = 1) | 15.7% (n = 11) |
| Diabetes | 11.1% (n = 1) | 21.7% (n = 18) | 7.1% (n = 1) | 24.6% (n = 17) |
| Years Post IRR* | n.a. | 5.5 (±4.93) | 4.07 (±2.70) | 5.8 (±5.24) |
| Total Radiation dose (Gy) | n.a. | n = 85 | n = 15 | n = 70 |
| < 36 | | 7.1% (n = 6) | 6.7%(n = 1) | 7.1% (n = 5) |
| 45–54 | | 1.2% (n = 1) | 0% | 1.4% (n = 1) |
| 55–63 | | 7.1% (n = 6) | 20% (n = 3) | 4.3% (n = 3) |
| 64–72 | | 81.2% (n = 69) | 73.3% (n = 11) | 82.9% (n = 58) |
| 73–81 | | 3.5% (n = 3) | 0% | 4.3% (n = 3) |
| Symptom severity | | | | |
| Frequency (>8 voids) | 28.6% (n = 2/7) | 34.2% (n = 27/79) | 57.1% (n = 8/14) | 29.2% (n = 19/65) |
| 9–11 voids | 50% (n = 1) | 66.7% (n = 18) | 50% (n = 4) | 73.7% (n = 14) |
| 12–14 voids | 50% (n = 1) | 22.2% (n = 6) | 25% (n = 2) | 21.1% (n = 4) |
| > 14 voids | 0% | 11.1% (n = 3) | 25% (n = 2) | 5.3% (n = 1) |
| Urgency | 0% | 35.3% (n = 30/85) | 53.3% (n = 8/15) | 31.4% (n = 22/70) |
| Hematuria | 0% | 11.8% (n = 10/85) | 60% (n = 9/15) | 1.4% (n = 1/70) |
| Gross hematuria | | 50% (n = 5) | 55.6% (n = 5) | 0% |
| Nocturia | 66.7% (n = 6/9) | 78.8% (n = 67/85) | 100% (n = 15/15) | 74.3% (n = 52/70) |
| Once/night | 16.7% (n = 1) | 17.9% (n = 12) | 20% (n = 3) | 17.3% (n = 9) |
| Multiple times/night | 83.3% (n = 5) | 82.1% (n = 55) | 80% (n = 12) | 82.7% (n = 43) |
| Urinary incontinence | 0% | 9.4% (n = 8/85) | 20% (n = 3/15) | 7.1% (n = 5/70) |
| Bladder spasm | 0% | 5.9% (n = 5/85) | 20% (n = 3/15) | 2.9% (n = 2/70) |

* Averages are given (± SD). BMI: body mass index; IRR: irradiation.

This study identified altered levels of three fibrotic markers (PAI 1, TIMP1 and TIMP2), as well as two vascular markers (HGF and VEGF-A). While we classified these proteins as pro-fibrotic or vascular, there is crosstalk between these pathways. In normal tissue, the homeostasis of extracellular matrix (ECM) is maintained through a balance between production and breakdown of collagens. Disruption of this homeostasis is observed when increased production of ECM is necessary to restore damaged tissue. Chronic tissue insult (e.g. hypertension, liver disease or diabetes), and subsequent repetitive need for tissue repair, can result in excessive accumulation of ECM, leading to tissue fibrosis. In the bladder, fibrosis thickens the bladder wall and alters its urodynamic compliance [23]. We identified increased levels of three proteins (PAI 1, TIMP1 and TIMP2) that can support a pro-fibrotic environment.

Plasminogen activator inhibitor 1 (PAI 1), or Serpin E1, inhibits the activity of urokinase-type/tissue type plasminogen activator (uPA/tPA) and plasmin. Hereby, PAI 1 indirectly inhibits the activation of plasminogen-dependent matrix metalloproteinases, such as MMP2 and MMP9, resulting in hindrance of the proteolysis of ECM and stimulation of fibrosis. As such, sustained high levels of PAI 1 are associated with fibrosis in various organs, e.g. heart, skin, liver, kidney and lung [10, 24]. Through the inhibition of uPA/tPA, PAI 1 also inhibits

**Table 3. Protein function of fibrotic and angiogenic proteins.**

| | Protein Function | Disease implications |
|---|---|---|
| **Fibrotic Markers** | | |
| **PAI-1**<br>Plasminogen Activator Inhibitor 1 | Enhances ECM deposition by blocking MMP activation<br>Stimulates clot formation by inhibiting fibrinolysis [10] | Implicated in many pathologies including tissue fibrosis, obesity, cardiovascular disease [10, 11] |
| **TIMP-1**<br>Tissue Inhibitor of Matrix Metalloproteinase 1 | Restricts ECM proteolysis by inhibiting activity of MMPs, including MMP-1, MMP-3 and MMP-9 [12] | Elevated in pulmonary, myocardial and hepatic fibrosis, and hepatitis C liver disease [13–17] |
| **TIMP-2**<br>Tissue Inhibitor of Matrix Metalloproteinase 2 | Inhibits activity of MMPs, including MMP-2 [12] | Elevated in hepatic fibrosis and hepatitis C liver disease [13, 17] |
| **Vascular Markers** | | |
| **HGF**<br>Hepatocyte Growth Factor | Stimulates angiogenesis by promoting proliferation, migration and survival of endothelial cells<br>Inhibits fibrosis<br>Regulates inflammation<br>Tissue regeneration [18, 19] | Decreased in COPD<br>Suggested as treatment to: inhibit fibrosis, enhance tissue regeneration, treat ischemic diseases [19–21] |
| **VEGF-A**<br>Vascular Endothelial Growth Factor A | Important regulator of vascular health by stimulating development and maintenance of blood vessels [22] | Therapeutic target to stimulate or suppress angiogenesis (e.g. cancer, diabetic retinopathy) [22] |

fibrinolysis hereby stimulating blood clot formation [25]. Thus PAI 1 could have a dual role in RC: 1) promote bladder fibrosis, 2) stimulate blood clot formation to help repair damaged blood vessels and arrest bleeding. PAI 1 levels are dependent on various factors, such as body composition and time of day; PAI 1 levels follow a circadian rhythm with highest levels in the morning [10]. In this study, we collected urine samples at random time points of the day, which can explain the large variability observed between samples.

Similar to PAI 1, tissue inhibitor of matrix metalloproteinase-1 and 2 (TIMP1 and TIMP2) can each block various MMPs from degrading the ECM [12, 26]. TIMP1 is used, in conjunction with 2 other proteins, in the Enhanced Liver Fibrosis test (ELF$^{TM}$; Siemens Healthineers), which determines a patient's risk for liver fibrosis [27]. Although elevated levels of TIMP1 in serum is an indicator of liver fibrosis, there is conflicting data from studies using pre-clinical models on how and if TIMP1 functionally contributes to tissue fibrosis [26]. Similar to liver fibrosis, our study suggests that TIMP1 could be a marker for bladder fibrosis/RC. TIMP2 has not been widely studied, though in a chemical model of liver fibrosis TIMP2 was shown to be pro-fibrotic [28]. It is feasible that TIMP1 and TIMP2 have a synergistic effect on fibrosis by inhibiting MMPs together.

As we previously reported and discussed, we identified elevated levels of hepatocyte growth factor (HGF) and vascular endothelial growth factor A (VEGF-A) in urine of irradiated patients with RC diagnosis or symptoms [8]. VEGF-A is a master regulator of vascular homeostasis and angiogenesis. It can be produced by a large variety of cells and primarily binds its receptors on vascular and lymphatic endothelial cells to promote vasodilation, angiogenesis, and vascular permeability and homeostasis [29]. VEGF-A is increased in response to hypoxia, thus elevated levels of VEGF could indicate the presence of hypoxic tissue in the bladder. Like VEGF, HGF stimulates angiogenesis by inducing cell migration, proliferation, survival and morphogenesis [20]. However, HGF has also been shown to protect against pulmonary fibrosis. Thus, HGF could potentially play a dual role in RC whereby it promotes angiogenesis and is protective against fibrosis.

Most of the inflammatory proteins were below detectable range of the assay. The absence of pro-inflammatory proteins in the urine of RC patients is striking, given that inflammatory

proteins are elevated in urine samples of patients with other bladder conditions, including interstitial cystitis, bacterial cystitis, bladder cancer, and urinary tract infections [6, 30, 31]. The cytokines IL-1a, IL-6, IL-8 and MIP-1a were detectable in one or two RC patients with gross hematuria and high symptom scores. Likewise, pre-clinical models of RC have not demonstrated a chronic inflammatory response to radiation [32–36]. One study did identify elevated levels of macrophage migration inhibitory factor (MIF) in patients with RC. Though MIF levels were compared to healthy individuals without radiation therapy history and thus we cannot conclude that these changes were radiation-induced or specific to patients with RC [37]. Thus, our findings suggest that inflammation may not play a prominent role during RC disease progression. Rather, bladder inflammation in RC patients may be indicative of end stage disease.

This study has several limitations. First, the proteins analyzed in this study were limited to the analytes available for use on the Multiplex Luminex system; other inflammatory, fibrotic or vascular proteins could be increased in the urine as well. Second, urinary protein levels might not fully reflect bladder pathology; no detectable change in the urine does not mean that analytes are not altered in the bladder itself. Third, the data is based on diagnosis and symptoms at time of urine collection; patients without symptoms at time of sample collection could develop RC-related symptoms at a later time point. This means that ongoing bladder changes could already demonstrate higher proteins levels in the urine without obvious functional bladder changes (e.g. hematuria, nocturia). Finally, we acknowledge that statistical significance at 5% level was not obtained in all comparisons. The Welch T-test was used to correct for high variance in the data, which compromises the p-value. Comparing protein changes over time within one individual through a longitudinal study would likely yield less variability in the data.

## Conclusion

This is the first study that extensively explored changes in inflammatory, fibrotic and vascular proteins in urine of PCa survivors with radiation therapy history as a means to understand radiation damage to the bladder. Our data suggests that RC and associated symptoms are primarily driven by fibrosis and vascular damage and/or remodeling. Lack of changes in inflammatory cytokines suggests that inflammation is not a key characteristic of RC and thus should not be the primary target for treatment, unless for patients that are suffering from end-stage disease. Furthermore, many studies have shown that urinary-based biomarkers have high sensitivity and specificity in the diagnosis of bladder diseases (e.g. bladder cancer), thus the findings of this study provide initial evidence that urinary biomarkers might help predict, at an early stage, who will develop RC.

## Supporting information

**S1 File. Patient questionnaire–Spanish.** After providing written informed consent, all participants completed this survey to collect information on patient demographics, prostate cancer history, radiation therapy, and bladder health.
(PDF)

**S2 File. Patient questionnaire–English.** English translation of the patient questionnaire.
(PDF)

## Acknowledgments

We would like to acknowledge Elijah Ward for technical assistance during this study.

## Author Contributions

**Conceptualization:** Bernadette M. M. Zwaans, Heinz E. Nicolai, Michael B. Chancellor, Laura E. Lamb.

**Data curation:** Bernadette M. M. Zwaans.

**Formal analysis:** Bernadette M. M. Zwaans, Laura E. Lamb.

**Funding acquisition:** Bernadette M. M. Zwaans, Michael B. Chancellor, Laura E. Lamb.

**Investigation:** Bernadette M. M. Zwaans.

**Project administration:** Bernadette M. M. Zwaans.

**Resources:** Bernadette M. M. Zwaans, Heinz E. Nicolai, Michael B. Chancellor, Laura E. Lamb.

**Supervision:** Michael B. Chancellor, Laura E. Lamb.

**Visualization:** Bernadette M. M. Zwaans, Laura E. Lamb.

**Writing – original draft:** Bernadette M. M. Zwaans.

**Writing – review & editing:** Bernadette M. M. Zwaans, Heinz E. Nicolai, Michael B. Chancellor, Laura E. Lamb.

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
