## [Decision Letter · Decision Letter 0]

4 Aug 2020

PONE-D-20-20879

Prostate cancer survivors with symptoms of radiation cystitis have elevated fibrotic and vascular proteins in urine

PLOS ONE

Dear Dr. Lamb,

Thank you for submitting your manuscript to PLOS ONE. After careful consideration, we feel that it has merit but does not fully meet PLOS ONE’s publication criteria as it currently stands. Therefore, we invite you to submit a revised version of the manuscript that addresses the points raised during the review process.

A rebuttal letter that responds to each point raised by the academic editor and reviewer(s). You should upload this letter as a separate file labeled 'Response to Reviewers'. Since a number of separate issues were raised by both reviewers, please ensure that your response is comprehensive and adequately addresses the concerns raised by both reviews.A marked-up copy of your manuscript that highlights changes made to the original version. You should upload this as a separate file labeled 'Revised Manuscript with Track Changes'.An unmarked version of your revised paper without tracked changes. You should upload this as a separate file labeled 'Manuscript'.

We look forward to receiving your revised manuscript.

Kind regards,

Praveen Thumbikat

Academic Editor

PLOS ONE

Journal Requirements:

2. Please provide additional details regarding participant consent. In the ethics statement in the Methods and online submission information, please ensure that you have specified whether consent was informed.

3. Please note that PLOS does not permit references to “data not shown.” Authors should provide the relevant data within the manuscript, the Supporting Information files, or in a public repository. If the data are not a core part of the research study being presented, we ask that authors remove any references to these data.

4, In your Methods section, please provide additional information about the participant recruitment method and the demographic details of your participants. Please ensure you have provided sufficient details to replicate the analyses such as: a) the recruitment date range (month and year), b) a description of how participants were recruited, and c) descriptions of where participants were recruited and where the research took place.

5. Please provide a sample size and power calculation in the Methods, or discuss the reasons for not performing one before study initiation.

6. Please include additional information regarding the survey used in the study and ensure that you have provided sufficient details that others could replicate the analyses. For instance, if you developed a questionnaire as part of this study and it is not under a copyright more restrictive than CC-BY, please include a copy, in both the original language and English, as Supporting Information.

7. To comply with PLOS ONE submission guidelines, in your Methods section, please provide additional information regarding your statistical analyses. For more information on PLOS ONE's expectations for statistical reporting, please see https://journals.plos.org/plosone/s/submission-guidelines.#loc-statistical-reporting.

8. We note that you have included the phrase “data not shown” in your manuscript. Unfortunately, this does not meet our data sharing requirements. PLOS does not permit references to inaccessible data. We require that authors provide all relevant data within the paper, Supporting Information files, or in an acceptable, public repository. Please add a citation to support this phrase or upload the data that corresponds with these findings to a stable repository (such as Figshare or Dryad) and provide and URLs, DOIs, or accession numbers that may be used to access these data. Or, if the data are not a core part of the research being presented in your study, we ask that you remove the phrase that refers to these data.

Reviewers' comments:

Reviewer's Responses to Questions

**Comments to the Author**

1. Is the manuscript technically sound, and do the data support the conclusions?

Reviewer #1: Partly

Reviewer #2: Yes

2. Has the statistical analysis been performed appropriately and rigorously? 

Reviewer #1: No

Reviewer #2: Yes

3. Have the authors made all data underlying the findings in their manuscript fully available?

Reviewer #1: Yes

Reviewer #2: Yes

4. Is the manuscript presented in an intelligible fashion and written in standard English?

Reviewer #1: Yes

Reviewer #2: Yes

5. Review Comments to the Author

Reviewer #1: Authors have sought to use urine analysis to investigate biomarkers for fibrosis in the bladder wall of radiation cystitis patients. Following concerns were noted

1) Experiments are predicated on the assumption that urinary elevation of (PAI 1, TIMP1 and TIMP2) proteins can serve as liquid biopsy for fibrosis. The underlying assumption is missing support from either urodynamic data or bladder wall histology to support a causal relationship between increased levels of three proteins (PAI 1, TIMP1 and TIMP2) and a pro-fibrotic environment in RC patients with hematuria.

2) Since hematuria is not unique to RC, authors should qualify the specificity of their findings with respect to other disorders exhibiting hematuria.

3) When authors claim that urinary proteins are stable for up to one year at room temperature with norgen preservative, what is the rationale for exposing urine samples to freeze thaw cycle before analysis. Did the authors consider analyzing urine samples directly without freeze thaw to rule out whether extensive processing of urine samples containing norgen preservative followed by centrifugation, freezing at -80 and thawing contributes to the striking absence of pro-inflammatory proteins in the urine of RC patients. Or is the lack of detection of urinary inflammatory proteins due to protein degradation during interval between collection and analysis.

4) The claim "inflammation is not the driving factor of the disease" in patients is speculative and not backed by any supportive evidence.

5) Is it correct that hematuria was seen in only 5 of the RC patients as per table 1. It was hard to link the number in table 1 with fig. 2 scatter plot for hematuria.

6) Are the p values corrected for multiple testing

7) More details of RC diagnosis needs to be added as "decreased maximal bladder capacity as measured during cystoscopy" does not make sense

Reviewer #2: This is in interesting well written paper looking at urinary levels of various urinary proteins to determine possible areas of abnormality to guide future therapy.

Page 9 line 59 - surely this should read " urinary diversion with or without cystectomy" not vice versa?

Page 10 line 73 - should read 'but are' not 'but is'

Page 11 line 85 - why are the patients from Chile only?

Page 11 line 90 - what was the maximal bladder capacity in no radiation, radiation with no radiation cystitis and radiation cystitis. The definition of radiation cystitis is vague and needs clarity.

Page 11 line 98 - what symptoms cires were used? It looks very much that this is a 'home made' symptom score. What was the reason for using this as opposed to the many validated and international recognised symptom scores such as IPSS and ICIQ OAB

Page 11 line 99 - likewise on what basis was is 10 selected as a cut off for the severe symptoms?

Page 11 line 105 - whta is th eevidence that storage/freezing and shipping didn't affect urinary protein levels?

Page 12 line 115 - how do the protein levels in those with IC compare to those with RC?

Page 13 Table - please add an addendum defining the markers and explaining their putative actions

Page 15 Table 2 - do you have urodynamic parameters or a frequency/volume chart to help with your definitions they are all somewhat vague?

Page 15 Table 2 - I don't think GU spasm is a recognised term - please can you define.

Page 21 line 281 - perhaps the samples would have been better collecting all at the same time ie 1st void of the day or 0900? Please comment on this and other factors that may affect urinary proteins - diet/hydration/medications - how can (and should they be) controlled for ?

Page 22 conclusions - Is significance the correct test for relevance of urinary proteins/biomarkers? Could there be a threshold phenomena - please discuss.

6. PLOS authors have the option to publish the peer review history of their article (what does this mean?). If published, this will include your full peer review and any attached files.

Reviewer #1: No

Reviewer #2: No

---

## [Author Response · Author response to Decision Letter 0]

2 Oct 2020

Thank you for the constructive feedback. We have made changes to the improved manuscript and hope that this is now acceptable for publication.

Reviewer #1: Authors have sought to use urine analysis to investigate biomarkers for fibrosis in the bladder wall of radiation cystitis patients. Following concerns were noted

1) Experiments are predicated on the assumption that urinary elevation of (PAI 1, TIMP1 and TIMP2) proteins can serve as liquid biopsy for fibrosis. The underlying assumption is missing support from either urodynamic data or bladder wall histology to support a causal relationship between increased levels of three proteins (PAI 1, TIMP1 and TIMP2) and a pro-fibrotic environment in RC patients with hematuria.

While we agree with the reviewers point, we did not perform bladder wall histology on these patients. Taking biopsies of bladders with RC can result in severe complications for the patients and most patients will not consent for this. We did not collect urodynamic data for all participants. We attempt to determine maximal bladder capacity during cystoscopy in patients that are suspected to have RC, though due to the risk of tearing of fragile blood vessels, maximal bladder capacity is often not reached. We did compare frequency data with urinary levels of pro-fibrotic factors, but this yielded no significant results. 

2) Since hematuria is not unique to RC, authors should qualify the specificity of their findings with respect to other disorders exhibiting hematuria.

All patients with micro- or macrohematuria were studied with cystoscopy for confirmation of radiation cystitis and to rule out other causes of hematuria. We added this information to the manuscript to clarify. Thank you.

3) When authors claim that urinary proteins are stable for up to one year at room temperature with norgen preservative, what is the rationale for exposing urine samples to freeze thaw cycle before analysis. Did the authors consider analyzing urine samples directly without freeze thaw to rule out whether extensive processing of urine samples containing norgen preservative followed by centrifugation, freezing at -80 and thawing contributes to the striking absence of pro-inflammatory proteins in the urine of RC patients. Or is the lack of detection of urinary inflammatory proteins due to protein degradation during interval between collection and analysis.

For the urine samples both preservative and freezing were used for several reasons:

1. The preservative was used to allow for short term storage in the clinic and shipping of the samples at room temperature. This allowed for collection of a greater number of samples at the clinic before shipping, hereby minimizing the number of necessary shipments. This prevented possible freeze/thaw accidents due to hold up of samples at customs. Also, at the clinic they do not have direct access to a centrifuge to remove any debris in the urine, nor was there access to a -80C. If preservative is not used, urine samples need to be frozen down immediately after collection to avoid degradation of proteins/RNA.

2. Urine samples were collected over 3 years, and the preservative is only stable for up to 1 year.

3. To avoid inter-experimental variation, we ran all the urine samples at the same time. Thus, we needed to wait until all the samples were collected.

4. Multiplex Luminex plates are costly; thus we choose to only run full plates.

The process of using preservative and freezing the urine samples was also used by our group for our interstitial cystitis biomarker studies. In these studies we did find detectable and elevated levels of cytokines in the urine. In addition, several urine samples from patients with severe RC did contain high levels of cytokines, as we reported in this manuscript. Thank you.

4) The claim "inflammation is not the driving factor of the disease" in patients is speculative and not backed by any supportive evidence.

We agree with the reviewers and have altered the language in the manuscript. Thank you.

5) Is it correct that hematuria was seen in only 5 of the RC patients as per table 1. It was hard to link the number in table 1 with fig. 2 scatter plot for hematuria.

15 of the urine samples we collected were from patients that had been diagnosed with RC. Of these 15, 9 reported hematuria. Of the 9 RC patients with hematuria, 5 reported gross hematuria with blood clots. 

6) Are the p values corrected for multiple testing

No, the p-values are not corrected for multiple testing as only the means of 2 groups were tested in each one-way ANOVA test. Thank you.

7) More details of RC diagnosis needs to be added as "decreased maximal bladder capacity as measured during cystoscopy" does not make sense

The definition we used is the classical definition of RC: pale bladder mucosa and telangiectasia. The bladder has fragile vessels that bleed easily during the endoscopy. If possible, we tried to fill the bladder to its maximum capacity, but because of the danger of inducing more bleeding, this is often not feasible. Most patients with macrohematuria receive a bladder catheter in the emergency room before they are admitted in the department of Urology to remove blood clots and to irrigate the bladder. When a cystoscopy is performed afterwards, we find erosion of the bladder mucosa and edema related to the use of the catheter.

Reviewer #2: This is in interesting well written paper looking at urinary levels of various urinary proteins to determine possible areas of abnormality to guide future therapy.

Page 9 line 59 - surely this should read " urinary diversion with or without cystectomy" not vice versa? 

We apologize for the confusion. With urinary diversion we were eluting to the reconstruction of a neobladder. We have corrected this in the manuscript.

Page 10 line 73 - should read 'but are' not 'but is'. – this grammatical error was corrected.

Page 11 line 85 - why are the patients from Chile only? 

This study was done in collaboration with Dr. Heinz Nicolai, a urologist from the University of Chile in Santiago de Chile. He was responsible for patient recruitment and urine collection. Blinded urine samples were subsequently shipped to Beaumont, processed and analyzed. RC is considered an orphan disease and recruiting a large number of patients without introducing too many variables (e.g. ethnicity, radiation dose, treatment strategy, type of pelvic cancer, gender, age) is challenging. To minimize the number of variables, we recruited only men with a history of prostate cancer that received external beam radiation therapy at one location. Future studies will focus on analyzing urine samples from patients from different locations, and with different cancer history (e.g. cervical cancer, colorectal cancer).

Page 11 line 90 - what was the maximal bladder capacity in no radiation, radiation with no radiation cystitis and radiation cystitis. The definition of radiation cystitis is vague and needs clarity.

No urodynamic measurements were performed in all participants. As mentioned earlier, we attempt to determine the maximal bladder capacity in patients that are suspected to have RC and that receive a cystoscopy. However, due to the fragile nature of the bladder vasculature, reaching maximal bladder capacity is often not feasible. Therefore, we cannot provide you with maximal bladder capacity for the different groups.

As for the definition of radiation cystitis: this is the classical way in which RC is diagnosed using cystoscopy. Please see also our response to reviewer #1 (Comment #7).

Page 11 line 98 - what symptoms cires were used? It looks very much that this is a 'home made' symptom score. What was the reason for using this as opposed to the many validated and international recognized symptom scores such as IPSS and ICIQ OAB.

We developed this survey specifically for RC and have used it in a previous publication [Zwaans et al. 2016 (ref 8)]. We included the reference in the revised manuscript.

Page 11 line 99 - likewise on what basis was is 10 selected as a cut off for the severe symptoms? 

A score of 10 corresponds to the presence of two symptoms (hematuria, gross hematuria, incontinence, nocturia or GU spasm) occurring multiple times per day, or to 3 or more symptoms occurring at minimum once per week. We believe this is a clear indication of an underlying urological defect.

Page 11 line 105 - what is the evidence that storage/freezing and shipping didn't affect urinary protein levels?

Protein and nucleotides in urine samples are extremely vulnerable to degradation at room temperature and thus it is really important that urine samples are processed in a timely manner after collection. The standard operating procedure for urine collection and storage is: Spin down at low speed, aliquoting supernatant and storing at -80C. Here are several studies on urinary cytokines in which this process has been followed (PMID: 29891542, 20942931, 26119560). In the latter study (PMID 26119560) the researchers added a ‘homemade’ preservative to urine samples prior to storage at -80C to ensure protein stability was maximized. Urine samples were from patients with non-muscle invasive bladder cancer. They subsequently performed Multiplex Luminex assay, as we did in our study. Cytokines analyzed in this study included IL-2, IL-8, IL-6, IL-1ra, IL-10, IL-12[p70], IL-12[p40], TRAIL, and TNF-α. As an example, reported ranges for IL-6 were (0 – 2780 pg/ml) and for IL-8 (-12 – 2998 pg/ml).

For our study, we ensured that urine samples were only frozen once prior to protein analysis to minimize protein degradation due to freeze/thaw cycles.

Page 12 line 115 - how do the protein levels in those with IC compare to those with RC? 

We have not made a direct comparison between IC samples and RC samples of all the analytes reported in this study. Our group has reported elevated levels of IL-6, IL-8 and GRO (CXCL1) in patients with IC (with and without ulcers) in comparison to control urine (PMID: 29088231). In IC patients, IL-6 ranged from 1.4-4.4 pg/ml, IL-8 ranged from 20-54 pg/ml, and GRO ranged from 10-21.5 pg/ml. For RC samples in which IL-6 and IL-8 were detected, IL-6 levels were on average 2.1 pg/ml and IL-8 levels were on average 20 pg/ml. But it is important to note that these two cytokines were only detected in a small portion of samples. 

Page 13 Table - please add an addendum defining the markers and explaining their putative actions. A third table has been added with this information

Page 15 Table 2 - do you have urodynamic parameters or a frequency/volume chart to help with your definitions they are all somewhat vague?

No, we do not have urodynamic parameters available. We did collect frequency data that is documented in Table 2. In the early stages of RC, we often find an increase of their void frequency, later in an advanced state of RC, we clearly observe pollakiuria, nocturia and urgency.

Page 15 Table 2 - I don't think GU spasm is a recognized term - please can you define.

We changed this to Bladder spasm. We agree that the terminology of spasm can be inappropriate here since we generally observe the spasm in patients with RC that require a bladder catheter.

Page 21 line 281 - perhaps the samples would have been better collecting all at the same time i.e. 1st void of the day or 0900? Please comment on this and other factors that may affect urinary proteins - diet/hydration/medications - how can (and should they be) controlled for?

Collecting urine at the first void of the day could help in tightening the data for certain factors such as PAI-1. However, our goal is to identify urine biomarkers that are robust enough so they are not sensitive to the time at which urine was collected. In addition, we have 

Page 22 conclusions - Is significance the correct test for relevance of urinary proteins/biomarkers? Could there be a threshold phenomena - please discuss.

We agree that a threshold for urinary protein levels rather than a significance level could be more relevant with respect to identifying potential urinary biomarkers for RC. However, in this study, we believe that using a significance test is more appropriate. First, the goal of the study was to use urine samples as “liquid biopsies” to identify proteins that play a role in the disease progression and not per se to identify protein biomarkers. A follow-up study could look into which of these proteins could function as possible biomarkers. Second, to set a clear threshold we will have to increase the number of samples, especially for proteins with small concentration ranges, and collecting samples at same time of day to minimize data variability would be needed. 

There is a fair amount of inter-subject variability with respect to levels of the tested proteins in the urine. We believe that the best method to screen for the development of RC using urine, is to track the concentration of several factors over time within one individual. But this idea would need to be tested using a longitudinal study.

---

## [Decision Letter · Decision Letter 1]

14 Oct 2020

Prostate cancer survivors with symptoms of radiation cystitis have elevated fibrotic and vascular proteins in urine

PONE-D-20-20879R1

Dear Dr. Lamb,

We’re pleased to inform you that your manuscript has been judged scientifically suitable for publication and will be formally accepted for publication once it meets all outstanding technical requirements.

Kind regards,

Praveen Thumbikat

Section Editor

PLOS ONE

Additional Editor Comments (optional):

Reviewers' comments:

Reviewer's Responses to Questions

**Comments to the Author**

1. If the authors have adequately addressed your comments raised in a previous round of review and you feel that this manuscript is now acceptable for publication, you may indicate that here to bypass the “Comments to the Author” section, enter your conflict of interest statement in the “Confidential to Editor” section, and submit your "Accept" recommendation.

Reviewer #1: All comments have been addressed

2. Is the manuscript technically sound, and do the data support the conclusions?

Reviewer #1: Yes

3. Has the statistical analysis been performed appropriately and rigorously? 

Reviewer #1: Yes

4. Have the authors made all data underlying the findings in their manuscript fully available?

Reviewer #1: Yes

5. Is the manuscript presented in an intelligible fashion and written in standard English?

Reviewer #1: Yes

6. Review Comments to the Author

Reviewer #1: Authors have responded constructively to earlier comments. They should consider following recommendations

Since all the significant results in Fig.2 and 3 are linked with the symptom of hematuria and the urodynamic data is missing, authors should emphasize on hematuria symptom in the title and text of the manuscript .

7. PLOS authors have the option to publish the peer review history of their article (what does this mean?). If published, this will include your full peer review and any attached files.

Reviewer #1: No

---

## [Editor Report · Acceptance letter]

16 Oct 2020

PONE-D-20-20879R1 

Prostate cancer survivors with symptoms of radiation cystitis have elevated fibrotic and vascular proteins in urine 

Dear Dr. Lamb:

I'm pleased to inform you that your manuscript has been deemed suitable for publication in PLOS ONE. Congratulations! Your manuscript is now with our production department. 

Kind regards, 

on behalf of

Dr. Praveen Thumbikat 

Section Editor

PLOS ONE